# OpenReview forum: "IMPersona: Evaluating Individual Level LLM Impersonation"
_colmweb.org/COLM/2025/Conference — COLM 2025_

### Official Review · Reviewer_YRPf · 2025-05-08

**Rating:** 8
**Confidence:** 4
**Ethics Flag:** 1

**Summary:**

The paper proposes IMPersona, a framework for developing and evaluating language models tailored to simulate specific individuals. Development consists of supervised fine-tuning using text written by the impersonated individual and a hierarchical memory module. Evaluation covers stylistic and contextual conversation topics, designed to measure how good the LLM is at impersonating the communication style, background knowledge, and experiences of the individual. Human evaluators who know the impersonated individuals interact with the LLM to rate how human-like the conversation was, and to decide whether they were interacting with an LLM or the impersonated individual themself.
The experiments show that a personalized Llama-3.1-8b-instruct model can fool humans into believing they are interacting with someone they know 44% of the time, whereas closed-weight models only reach a 25% pass rate.

**Questions To Authors:**

Questions:
- In the Memory Module Design paragraph, are the (N=7) retrieved memories first-order?
- What LLM is used for memory management?
- Did not understand this passage: "in some cases, our study was limited by data scarcity, with only 20 samples collected from two different evaluators per impersonated individual." Does this refer to evaluation or training data? I thought it was training data because the paragraph is about dataset size and performance, but then I do not understand what it means to collect training samples from evaluators.
- Table 1: "P-values indicate statistical significance in regression analysis." What regression analysis?

Suggestions:
- Include more details on the memory module design, e.g., what prompts are used, how lower-order memories are selected.
- Use \citet{} when the reference is included in the sentence. E.g., "Xu et al. (2024a) used LMs [...]" instead of "(Xu et al., 2024a) used LMs [...]".
- Include a reference in the main body to the appendix with the LoRA hyperparameters.

Typos:
- Figure 1 caption lacks a period at the end.
- Table 5 caption is incomplete.
- Appendix 7 -> Figure 7.

**Reasons To Accept:**

- The paper proposes a novel setup for training and evaluating LLMs personalized at the individual level, showing that even lightweight fine-tuning combined with a memory mechanism is enough to fool people into believing that an LLM is someone they know. The paper demonstrates how personalized LLMs can be used maliciously and points to some promising ways to mitigate that.
- The results demonstrate the effectiveness of the proposed fine-tuning scheme and memory mechanism, which enabled a Llama-3.1-8b-instruct model to be a better impersonator than much stronger, closed-weight models.
- The evaluation setting is convincing, using human evaluators to rate the humanness of conversations.

**Reasons To Reject:**

- Fine-tuning is conducted on a single model, so it is unclear what the effect would be in other model families or scales.
- Given the nature of the task and evaluation, the paper could provide more details on who were the impersonated humans, the evaluators, and how they were recruited. This seems like a very challenging process, given that the evaluators must know the impersonated humans, so it would be interesting to know more about how the authors achieved this.

---

> ### Author Response · Authors · 2025-05-31
>
> Thank you for the thoughtful and constructive review. We are encouraged by your positive assessment of our work, particularly your recognition of the novelty of our setup, the effectiveness of lightweight fine-tuning combined with a memory mechanism, and the realism of our evaluation design. We appreciate your acknowledgment of the potential implications, both positive and concerning, of personalized LLMs, as well as your interest in the mitigation strategies we propose. Below, we address your concerns and questions in more detail.
>
> ### **1. On the Use of a Single Model Family for Fine-tuning:**
> We agree that broader model diversity would be valuable and appreciate your pointing this out. As noted in our limitations section, we were constrained by available compute resources, particularly for larger models like the 70B class. Furthermore, expanding to multiple model families would significantly increase the number of human evaluators required to maintain statistical significance, given our evaluation protocol. Given these constraints, we prioritized evaluating performance across varying dataset sizes, as we believe this axis has broader implications in that if high impersonation performance can be achieved with small training datasets, this would pose a significant safety concern. That said, we did conduct preliminary experiments comparing other model families of similar size (e.g., Qwen-2.5-7B-Instruct vs. LLaMA-3.1-8B-Instruct), as reported in Section 4.
>
> ### **2. Recruitment of Human Evaluators:**
> As described in Section 3, each impersonated individual trained a model on their own data. For evaluation, they recruited up to 12 individuals who knew them personally: ranging from acquaintances to family members to ensure a baseline level of familiarity. This setup enabled us to simulate realistic human interactions and provide meaningful assessments of impersonation quality. We appreciate your suggestion and will update the manuscript to include a more detailed description of this process in the final version.
>
> ### **3. Further Clarifications:**
> **Memory Retrieval:** The seven retrieved memories used during interaction include a combination of top-ranked, first-order, and second-order memories.
>
>
> **Memory Management LLM:** We used GPT-4o to manage memory retrieval and selection, as we found that a strong reasoning model was essential for effective memory coordination.
>
>
> **Data Scarcity and Sample Size:** The “20 samples collected from two evaluators” refers to evaluation conversations, not training data. Specifically, in our analysis of scaling performance with dataset size (Figure 4), we had fewer evaluation interactions per condition (~20 conversations per impersonated individual across two evaluators) compared to the ~100 conversations used for the main setups reported in Table 1. This resulted in higher variance in some conditions, which we will make clearer in the text.
>
>
> **Regression Analysis (Table 1):** The regression model was an ordinary least squares (OLS) analysis with humanness ratings as the dependent variable. Predictors included impersonation type (coded as binary variables corresponding to the nine setups in Table 1), participant familiarity (closeness), texting frequency, prior AI exposure, and related covariates. We will ensure the regression methodology is clearly described in the final version.
>
> Thanks again for your thoughtful review, and for pointing out details that weren’t clear from your perspective! We will be sure to update the paper to reflect these details in the final version.

---

> > ### Comment · Reviewer_YRPf · 2025-06-02
> >
> > Thank you for the response!
> > I am maintaining my positive assessment of the work, and suggest that the authors include the clarifications in the camera-ready version.

---

### Official Review · Reviewer_WMAf · 2025-05-13

**Rating:** 7
**Confidence:** 4
**Ethics Flag:** 1

**Summary:**

An interesting problem is how well LLM systems can impersonate specific individuals rather than general human characteristics. The authors create a framework to evaluate LLMs' ability to write in a person's style and to recognize and utilize a person's background. Findings indicate that models could fool humans 44% of the time. The more that users text (unclear whether this is a general amount of texting or with a specific user), the easier it is for them to identify them as an impostor. Finally, the authors highlight/propose the most effective detection methods their participants utilize.

**Reasons To Accept:**

Your results are very interesting! In context, learning does a bad job of matching a person's texting style. Fine-tuning with just a few examples captures a person’s texting style more than in-context learning. Fine-tuned models create flow problems, aka changing the subject of the conversation. Increasing dataset size increases performance (although I’d love to know to what extent? Is there an upper limit or diminishing returns?) Your memory module is very clever and I like how you evaluated it against dense retrieval.

Page 3 - Human participants
This section is concise and packed with information. You did a great job listing the numerical details of your study and mentioning the IRB.

**Reasons To Reject:**

Overall, many of these are minor and not major weaknesses, but they are important to improve the paper.

There is a lack of clarity and detail in the experimental setup and methodology. For example, how long were sessions? Was the metadata analysis in 5.1 anecdotal, annotated, or LLM-generated, and if annotated, whether it was single-annotator or multi-annotator with agreement scores? What were the style and contextuality questions? Show your experiment from the pov of the subjects.

There is incomplete or unclear reporting of statistical analysis and results. Tables need clarifying details and better explanations. The regression analysis mentioned was not included in the paper in detail.

More actionable advice regarding these weaknesses is addressed below:
Page 2 - Introduction
“(less than $5 of compute on together.ai)” This needs context. More explanation on what this 5$ was spent on.

Page 2 – Related Work:
The sentence beginning "Turing Test with LMs Significant research..." uses the incorrect citation style. If you're using LaTeX, be sure to differentiate between \cite{} and \citet{}. In this case, because the citation serves as a noun within the sentence, you should use \citet{} to avoid parenthetical formatting.

Typo -> “Our methodologyd iffers” should be “methodology differs”

Page 3 - Stylistic + Contextual Prompts
“So to further analyze imitation quality, we collect additional metadata on reasoning for participant decisions, classifying between errors on style, context, and conversation flow.” it is unclear what this metadata looks like. Since you brought it up, please explain it.

“Participants struggled to identify the interviewer even in cases of close relationships (friends and siblings) if their texting interaction was infrequent.” Does this mean that the participants have low text interaction in general or with the specific interviewer / impersonator? It is unclear to the reader please clarify.

I want there to be a clearer experiment set up (not models and prompting / ft methods—those are excellent), but rather what the human is doing at the computer. How long are the messaging sessions? What are the questions asked before starting the chat? What are the questions asked after the chat? Humanness is a Likert scale question. Are Style and Cont? Who are the people answering the style cont questions? Are they the person in the chat room or the person being impersonated?

Page 6 - Table 1
This table is confusing. The caption of this table should better inform the reader of the content. For example:
What do the p-values mean? What do they correspond to? What is the significance of them all being zero? “Significance in regression analysis.” What regression analysis? You bring this up again on page 7. What are you regressing? What are your independent/dependent variables? Are you considering interaction variables, etc? Be explicit, show us exactly what you did.
You specify that Humannes is a 1-7 scale, but don’t specify for styl and cont?

Page 7-8  5.1 Analysis of Behavior
Your insights here are interesting, but you lack discussion on how you came by these insights. For example, are these discoveries anecdotal? As you, the researcher, read through the messages, are you presenting your overall impression of the strategies and their effectiveness? Did you prompt the LLM with the chat logs to identify these strategies? Did you annotate the chats with discrete categories and annotation guidelines? In the case that you did annotations, did a sole annotator do them? Did multiple? What are the agreement scores? Taking these steps to show how you identified these insights will give validity to your findings.

Page 8
“This capability is particularly concerning given emerging research showing LLMs being used to enhance phishing attacks with real-time contextual information.” needs citation

---

> ### Author Response · Authors · 2025-05-31
>
> Thank you very much for your thorough, thoughtful and constructive review. We greatly appreciate the time and care you took in providing detailed feedback, and we’re especially grateful for your recognition of both the strengths and implications of our findings. It seems that your main concerns center around minor clarifications in our presentation of methods and results. Below, we address each of your points in turn.
>
> ### **1. General Clarifications:**
> > For example, how long were sessions?
>
> Each session lasted 3 minutes, which is noted in Section 3, paragraph 2.
>
> > Was the metadata analysis in 5.1 anecdotal, annotated, or LLM-generated, and if annotated, whether it was single-annotator or multi-annotator with agreement scores?
>
> The analysis was annotated by humans, and was done by a single annotator. We will add this.
>
> > What were the style and contextuality questions? Show your experiment from the pov of the subjects.
>
> These are shown in Table 3 of the Appendix, with participant interface details provided in Figures 10 and 11.
>
> > More explanation on what this 5 was spent on
>
> The $5 refers to the approximate cost to fine-tune a model on the B500 dataset using LLaMA-3.1-8B-Instruct, illustrating the low barrier to launching personalized impersonation attacks. We will revise this line to provide more context.
>
> >  The regression analysis mentioned was not included in the paper in detail.
>
> This is a great point, and we will add this into our methodology section in detail in the final version for sure, as we only briefly describe it in the paper. For clarity, the regression model was an ordinary least squares (OLS) analysis with humanness ratings as the dependent variable. Predictors included impersonation type (coded as binary variables corresponding to the nine setups in Table 1), participant familiarity (closeness), texting frequency, prior AI exposure, and related covariates.
>
> > “So to further analyze imitation quality, we collect additional metadata on reasoning for participant decisions, classifying between errors on style, context, and conversation flow.” it is unclear what this metadata looks like. Since you brought it up, please explain it.
>
> Thank you for pointing out the unclearness here! Specifically, participants indicate the reasoning behind their choices using a predefined menu [See Figure 11 in the Appendix] classifying whether their judgment was influenced primarily by stylistic imitation, contextual alignment, or conversation flow (e.g., coherence, naturalness). They also have the option to provide open-ended verbal feedback to elaborate on their selections. Examples can be found in Appendix Figures 14-17. We will be sure to make this clearer in the final version! Thanks for the suggestion!
>
> > “Participants struggled to identify the interviewer even in cases of close relationships (friends and siblings) if their texting interaction was infrequent.” Does this mean that the participants have low text interaction in general or with the specific interviewer / impersonator? It is unclear to the reader please clarify.
>
> It is with regard to the specific interviewer / impersonator. We will revise this line to emphasize this.
>
> > What are the questions asked before starting the chat? What are the questions asked after the chat? Humanness is a Likert scale question. Are Style and Cont? Who are the people answering the style cont questions? Are they the person in the chat room or the person being impersonated?
>
> All questions and interface screenshots are included in Figures 10 and 11 of the Appendix. Due to space constraints, these were placed there, but we agree they are central to understanding the experiment and will reference them more directly. To clarify: Style and Contextual refer to the prompt type initiating the conversation, not additional Likert-style ratings. Each conversation was seeded with either a style-oriented or context-oriented prompt, and aggregate scores are grouped by this initial prompt category.
>
> We greatly appreciate your very thorough and precise feedback, and hope this addresses your concerns! We think your points were extremely constructive and we’ve revised the improved the paper in several places based upon your suggestions.

---

> > ### Comment · Reviewer_WMAf · 2025-06-02
> >
> > Thanks for the response. I will keep my score (it is high already). Great work.

---

### Official Review · Reviewer_jt2C · 2025-05-15

**Rating:** 7
**Confidence:** 4
**Ethics Flag:** 1

**Summary:**

This paper presents IMPersona, a framework for evaluating the ability of language models to impersonate specific individuals in conversational settings. The authors combine supervised fine-tuning (via LoRA), in-context learning, and a hierarchical memory retrieval system. They evaluate impersonation performance through a modified Turing test, where human participants interact with either a real person or an impersonated version generated by an LM, and then judge whether the conversational partner was human.

The study includes over 100 participants and 12 impersonated individuals. Results show that fine-tuned models with memory integration achieved a 44.44% deception rate, compared to 25.00% for the best prompting-only approach. Performance was analyzed across multiple dimensions including stylistic fidelity, contextual accuracy, and data size. The authors also examine detection strategies, model failure modes, and ethical risks related to impersonation.

**Questions To Authors:**

* Can you clarify the extent to which your memory module performs higher-level synthesis versus surface-level retrieval? While the hierarchical memory system is described in detail, it’s still somewhat ambiguous whether the impersonator model integrates abstract personality traits or primarily repeats recalled events.
* You report that participants were familiar with the impersonated individuals and include closeness ratings in the appendix. Could you provide a breakdown of impersonation success rates by familiarity level (e.g., family vs. acquaintance)? This would help quantify how well the models impersonate individuals known closely to the evaluator
* Why were prompting baselines limited to hand-crafted examples? Did you consider using soft prompting, prefix tuning, or frameworks like DSPy for automated prompt optimization? Without such comparisons, it is difficult to conclude that prompting is fundamentally less effective than fine-tuning.
* Your evaluation uses “pass rate” and “humanness ratings” as primary metrics. How do you distinguish between models that sound generally human versus those that convincingly replicate a specific individual?
* You note that contextual probing (e.g., questions about current events) was more effective than explicit adversarial prompts. How often were these strategies used across participant subgroups, and did their usage correlate with familiarity level or AI experience?
* Given that your participant pool skewed toward young, technically savvy individuals, how might the deception rates change with a more demographically diverse sample? Would older or less AI-experienced participants be more or less susceptible?

**Reasons To Accept:**

* The paper introduces a structured and reproducible framework for evaluating individual-level impersonation using conversational interactions, which goes beyond prior work focused on generic human vs. AI discrimination.
* The hierarchical memory module is clearly described and compared against a dense retrieval baseline, showing consistent improvement in contextual accuracy.
* The study demonstrates that effective impersonation can be achieved with as few as 500 message examples, indicating a low data threshold for these capabilities. They also run ablations showing clear scaling trend.
* The evaluation includes controlled testing conditions, participant metadata (e.g., texting frequency, familiarity), and multiple model variants, providing a robust comparison across configurations.
* Ethical considerations are addressed directly, with discussion of both potential harms (e.g., deception, phishing) and possible beneficial applications (e.g., education, simulation).

**Reasons To Reject:**

* While the paper provides a detailed description of a hierarchical memory system that involves abstraction and consolidation, the term memory integration is used inconsistently. It would help to more explicitly clarify the distinction between surface-level retrieval and higher-level synthesis of identity-relevant knowledge throughout the paper
* The paper states that participants were required to have some prior familiarity with the impersonated individuals, and Appendix materials include self-reported closeness ratings and texting frequency. However, the results do not break down impersonation success by relationship type (e.g., family member vs. acquaintance), making it difficult to assess how well the models perform under stronger familiarity conditions. The claim that even close contacts were deceived would be more compelling with performance data stratified by relationship closeness
* Prompting baselines rely on hand-crafted examples, which may underperform relative to optimized methods such as soft prompting or automated prompt tuning frameworks (e.g., DSPy). This limits the strength of the baseline comparisons.
* While the paper is focused on individual-level impersonation, the main evaluation metric—whether the model was mistaken as human—does not isolate whether the model accurately mimicked a specific individual versus simply sounding generically human. Although contextual prompts and familiarity ratings aim to support person-specific evaluation, the reported results do not fully disentangle general human-likeness from identity-level fidelity. This limits how confidently one can interpret impersonation success as identity-accurate rather than just plausibly human.

---

> ### Author Response · Authors · 2025-05-31
>
> Thank you for your thoughtful and constructive review. We are encouraged by your recognition of the novelty and rigor of our work, including the “structured and reproducible framework for evaluating individual-level impersonation,” the “clearly described” hierarchical memory module, and the detailed experimental design with participant metadata and multiple model variants. We also appreciate your acknowledgment of our ethical considerations and discussion of both potential risks and beneficial applications. Below, we address each of your concerns in turn:
>
> ### **1. Clarification on Memory Integration:**
> Our memory module primarily supports content-level grounding by surfacing causally related, event-specific memories organized through the hierarchical memory system. Its core function is not to shape the stylistic form of a response, but rather to retrieve relevant personal experiences or factual details that the impersonator model can use to construct contextually rich replies. For instance, in a conversation about favorite college classes, the memory system might retrieve details such as specific courses or classmates, which the impersonator model then integrates into a response that reflects the individual’s characteristic tone and narrative style.
>
> ### **2. Impersonation Success by Familiarity Level:**
> While Section 5 presents OLS regression analyses indicating that participant familiarity does not significantly affect humanness ratings, we agree that stratified results (e.g., by family member vs. acquaintance) could provide helpful context. We report below identification success rates based on closeness, which we will add to the appendix. As discussed in Section 6, even highly familiar participants can be deceived—particularly when they lack experience with AI-generated text or have limited exposure to the target’s digital communication style, evidenced below. Note that these numbers represent *identification* accuracy (how often participants identified AI as AI, and humans as humans across *all* settings).
>
> **Table: Accuracy by Tester Closeness to Experimenter**
>
>
> | Closeness Level                                            | Overall Accuracy (%) |
> |------------------------------------------------------------|-----------------------|
> | Between Acquaintances and Good Friends                     | 75.00                 |
> | Good Friends                                               | 73.88                 |
> | Core College Friends / Childhood Friend / Somewhat Close Family | 77.42            |
> | Best Friend                                                | 71.43                 |
> | Close Family                                               | 69.97                 |
>
>
> ### **3. Prompting Baselines and Prompt Optimization:**
> We agree that more sophisticated prompting strategies such as soft prompting or frameworks like DSPy could be promising. Our focus was on establishing a realistic and interpretable baseline for individual-level impersonation. Each prompting baseline was hand-crafted to incorporate known facts, style cues, and shared memories, already optimized towards the impersonation task. We think prompt optimizing ICL models could be a valuable direction for future work.
>
> ### **4. Clarification on Main Evaluation Metric:**
> We would like to clarify that our primary evaluation was not simply whether the model sounded human, but whether it convincingly impersonated a specific known individual. As noted in Section 1 and detailed in our evaluation protocol, participants interacted with someone they knew personally and were explicitly asked whether they believed the conversational partner was that person or an AI impersonator. The evaluation thus captures more than generic humanness; it reflects the model’s ability to replicate personal details, conversational quirks, and shared context.
>
> ### **5. Demographic Diversity:**
> As discussed in Section 7, our participant pool skewed toward younger and individuals with higher AI familiarity and experience. While this may limit generalizability, we argue it presents a stricter test, as participants with high familiarity with LLMs may be more attuned to their common communication patterns and thus more likely to apply effective probing strategies. Supporting this, our analysis shows a positive correlation between AI familiarity and detection success. In Section 7, we hypothesize that deception rates would likely be higher in a broader, less tech-savvy population.
>
> We hope this response clarifies points of confusion and supplements your understanding and perception of our work. Thank you again for your thoughtful review and questions!

---

> > ### Comment · Reviewer_jt2C · 2025-06-09
> >
> > Thanks so much for the replies! The additional details on the memory module and how it supports personal grounding helped clarify that point for me. One thing I’d encourage you to highlight a bit more explicitly in the limitations: since the memory module relies on retrieving detailed personal information and relationship-specific details, the impersonation capabilities depend heavily on access to such data — which may not always be available or ethically appropriate in real-world scenarios.
> >
> > Also appreciate the honest note re: prompting baselines — agree there’s lots of room for future work there.
> >
> > Overall, this response addressed my main questions.

---

> ### Comment · Area_Chair_55xq · 2025-06-05
> **Please acknowledge you've read author response**
>
> Dear reviewer,
>
> Could you please respond to let the authors know you have read their response?
>
> Thanks!
>
> -AC

---

### Official Review · Reviewer_3VQQ · 2025-05-27

**Rating:** 6
**Confidence:** 4
**Ethics Flag:** 1

**Summary:**

This paper uses supervised fine-tuning and retrieval augmentation to build an LM system for impersonating humans. With a backbone model like LLama 3.1 8B instruct, the system can mislead humans to believe it is a real human in 44.44\% of the interactions. The paper further studies potential detection methods and defense strategies against the LM impersonation risk.

**Reasons To Accept:**

1. The impersonation capabilities of LMs have great safety implications. This paper systematically benchmarks this risk with modern LMs by conducting real human-AI interaction experiments.
2. As explained in Sec2, the experiment is carefully designed, ruling out some potential confounding factors (e.g., typing seed)
3. The paper also studies potential detection methods, interesting findings are:
- Figure 4: prior AI exposure is a significant predictor of detection accuracy, while the closeness to participants is not
- Section 5.1: contextual probing is effective.

**Reasons To Reject:**

1. Participants are instructed to engage naturally with the conversation topics. This would discourage them from doing contextual probing, thus leading to poor detection accuracy. Could you report the frequency of contextual probing in their interactions?

2. regarding detection methods, I'd suggest evaluating general AI-generated text detectors.

3. potential overclaim: LMs readily engage in deceptive impersonation.  I don't see and quantitative analysis about deceptive impersonation.

---

> ### Author Response · Authors · 2025-05-31
>
> Thank you for your thoughtful review and feedback! We are encouraged that the reviewer recognizes the “great safety implications” of our novel study, our “carefully designed” experiments, as well as the interesting findings on predictors of success and mitigation strategies. We address and give relevant clarifications to your concerns below.
> ### **1. Frequency of intentional contextual probing:**
> As our goal was to simulate a naturalistic setting, where participants believed they were chatting with someone they knew, participants were not explicitly encouraged to test the identity of their interlocutor. In practice however, we observed in 45% of 20 randomly sampled conversations that participants engaged in a purposeful contextual probing. In real-world contexts, people typically do not suspect AI impersonation or engage in such contextual probing. Therefore, our evaluation seeks to measure baseline detection performance in realistic, everyday interactions, rather than the upper bound of human capability under suspicion or instruction, and we do not believe this to necessarily undermine our results.
>
>
> ### **2. Evaluating general AI-generated text detectors:**
> Thank you for this suggestion. While our primary focus was on human judgment, since humans are often the first line of defense in detecting impersonation, we agree that *effective* automated detectors could complement this process. However, in this work we chose not to systematically evaluate off-the-shelf detectors in this work for two key reasons:
> 1. [Main reason] Messaging data tends to be extremely short (avg ~35 characters per message), which greatly impairs detector performance. Prior work has shown that detection accuracy deteriorates significantly with shorter text lengths [2]. This is partly why leading AI text detectors like GPTZero only support inputs of over 250 characters, far exceeding the average message length of ~35 characters sent by our impersonator models.
> 2. Many existing detectors are trained on distributions of standard, well-known LLMs. Our fine-tuned models, by contrast, are trained on real human messaging data, substantially shifting their output distribution and reducing detector efficacy [1].
>
> To illustrate this point, we tested several example conversations from our study using GPTZero, a leading AI text detector. When inputting conversation histories from our fine-tuned LLaMA models (Appendix Figures 16 and 17), GPTZero classified the text as: *“We are highly confident this text is entirely human.”* In contrast, when we input conversation histories from prompted frontier models (Appendix Figures 14 and 15), GPTZero returned: *“We are uncertain about this document. If we had to classify it, it would likely be considered human.”* These results show the limited effectiveness of current detectors, particularly when applied to outputs from fine-tuned models optimized to mimic human messaging patterns.
>
> The lack of effective automated detectors for finetuned models further supports the fact that the individual-level impersonation problem is a critical safety issue for LLMs: one that has been largely unexplored in prior literature. The risk is further amplified by the increasing availability of real-world chat data on public platforms such as Discord and Reddit: something that we hope to highlight with our work in this paper.
> ### **3. Quantitative analysis about deceptive impersonation:**
> We appreciate your point regarding the need for quantitative support on deceptive impersonation. Our intention in highlighting "deceptive impersonation" was to demonstrate that LMs could be used to engage in behaviors that may deceive users: especially in familiar, personalized contexts, as exemplified in Figure 6. However, we intentionally avoided designing experiments that would mislead participants in ethically sensitive ways, as this could pose psychological risks violating IRB guidelines. As such, our focus is on characterizing the potential for such deception in realistic interactions, which we think is a significant implication of our results, rather than provoking deception explicitly.
>
>
> We hope this clears up any questions/confusion that you have about our work, and look forward to engaging in future discussion. Thanks again for your thoughtful review and questions!
>
> References:
>
> [1] Abdali, Sara, et al. "Decoding the AI pen: Techniques and challenges in detecting AI-generated text." KDD 2024.
>
> [2] Wang, Hao, Jianwei Li, and Zhengyu Li. "AI-generated text detection and classification based on BERT deep learning algorithm." arXiv:2405.16422 (2024).

---

> > ### Comment · Reviewer_3VQQ · 2025-06-08
> >
> > > In real-world contexts, people typically do not suspect AI impersonation or engage in such contextual probing.
> >
> > I don't agree with this. Because of phone fraud, people already do contextual probing when their "friend" talks about unusual topics, especially safety-related things.

---

> > > ### Author Response · Authors · 2025-06-09
> > >
> > > Yes, we do agree that contextual probing already occurs in everyday scenarios. That is why our study does not explicitly instruct participants to *not* contextually probe, but rather to engage with the impersonation *in a natural manner*: which means that if they do suspect AI, they can contextually probe it if they deem necessary, just like a real world scenario. Plenty of the participants end up asking personal questions on purpose to test the interlocutor, as mentioned above. Hope this clears up the intention behind our experiment!

---

### Decision · Program_Chairs · 2025-07-08

**Decision:**

Accept

**Comment:**

This paper presents a framework for evaluating the ability of language models to impersonate specific individuals in conversational settings, and they run extensive Turing-style tests, where participants are asked to converse with an agent who may either be a bot or a fellow human. They find that humans are fooled into thinking they are talking to a fellow human a signifiant portion of the time.

I expect this paper's method and results to be of broad interest at COLM, so I recommend for an oral presentation.

PROS
- well-written paper with interesting results
- user study is well-run, with lots of well-thought-out details to make the setup convincing
- framework is clearly described and reproducible by others

CONS
None that are significant

[Automatically added comment] At least one review was discounted during the decision process due to quality]